# Wnt Signaling Pathways: From Inflammation to Non-Melanoma Skin Cancers

**DOI:** 10.3390/ijms24021575

**Published:** 2023-01-13

**Authors:** Luca Di Bartolomeo, Federico Vaccaro, Natasha Irrera, Francesco Borgia, Federica Li Pomi, Francesco Squadrito, Mario Vaccaro

**Affiliations:** 1Department of Clinical and Experimental Medicine, Section of Dermatology, University of Messina, 98125 Messina, Italy; 2Department of Dermatology, University of Modena and Reggio Emilia, 41124 Modena, Italy; 3Department of Clinical and Experimental Medicine, Section of Pharmacology, University of Messina, 98125 Messina, Italy

**Keywords:** Wnt signaling, basal cell carcinoma, actinic keratosis, squamous cell carcinoma

## Abstract

Canonical and non-canonical Wnt signaling pathways are involved in cell differentiation and homeostasis, but also in tumorigenesis. In fact, an exaggerated activation of Wnt signaling may promote tumor growth and invasion. We summarize the most intriguing evidence about the role of Wnt signaling in cutaneous carcinogenesis, in particular in the pathogenesis of non-melanoma skin cancer (NMSC). Wnt signaling is involved in several ways in the development of skin tumors: it may modulate the inflammatory tumor microenvironment, synergize with Sonic Hedgehog pathway in the onset of basal cell carcinoma, and contribute to the progression from precancerous to malignant lesions and promote the epithelial-mesenchymal transition in squamous cell carcinoma. Targeting Wnt pathways may represent an additional efficient approach in the management of patients with NMSC.

## 1. Introduction

Wnt signaling is responsible for the regulation of different intracellular signal transduction pathways, which are essential for embryogenic development, cellular migration, polarization and differentiation as well as stem cell biology control and growth [1,2]. However, Wnt signaling network may also be related to immune-modulation and cancer development [2]; for this reason, the purpose of the present review is to summarize the most intriguing evidence about the role of Wnt signaling in cutaneous carcinogenesis, starting from pro-tumorigenic immunomodulation up to the development of non-melanoma skin cancers (NMSCs).

## 2. Wnt Signaling

Wnt signaling activation is related to the binding of Wnt ligands to a specific cell surface receptor which belongs to the Frizzled (Fzd) family [3], thus inducing the canonical (β-catenin-dependent) or non-canonical (β-catenin-independent) pathway [4] (Figure 1).

The Fzd receptor interacts with other coreceptors, such as the receptor-like tyrosine kinase (Ryk), the receptor tyrosine kinase-like orphan receptor 2 (Ror2) or the low-density lipoprotein receptor-related protein (Lrp)-5/6, with the consequent activation of Disheveled (Dvl) and the downstream pathways [3]. Generally, cytoplasmatic β-catenin is phosphorylated by a complex formed by glycogen synthase kinase 3β (Gsk3β), adenomatous polyposis coli (Apc), casein kinase 1α (Ck1α) and Axin; β-catenin phosphorylation leads to its degradation by proteasome [5,6]. β-catenin is stabilized when the degradation complex is inhibited by Dvl following the canonical Wnt pathway activation [2]. β-catenin translocation into the nucleus allows the interaction with transcription factors, such as lymphoid-enhancing factor/T-cell factor (Lef/Tcf) transcription proteins, thus promoting Wnt target genes transcription [7]. Canonical Wnt signaling role starts from stem cell differentiation until cell proliferation, both during embryogenesis and adult tissue homeostasis [8].

The non-canonical Wnt signaling pathways are divided into Wnt/Calcium (Ca^2+^) and Wnt/Planar cell polarity (PCP) pathways [3]. In particular, in the Wnt/Calcium (Ca^2+^) pathway, the activated Fzd-co-receptor-Dvl complex induces phospholipase C γ, which converts phosphatidylinositol 4,5-bisphosphate (PIP2) into diacylglycerol (DAG) and inositol 1,4,5-triphosphate (IP3), resulting in the increased release of intracellular Ca^2+^. The release of Ca^2+^ leads to the activation of calcium-dependent kinases, such as Ca^2+^-dependent phosphatase calcineurin (CaN), Ca^2+^-calmodulin dependent kinase II (CAMKII) or protein kinase C (PKC) [9]. The activation of CaMKII may induce the phosphorylation of TGFβ-activated kinase 1 (TAK1), resulting in Nemo-like kinase (NLK) activation. TAK1-NLK pathway stimulation antagonizes the canonical Wnt/β-catenin pathway [10]; on the other hand, CaN induces the translocation of nuclear factor of activated T-cells (NFAT) family proteins into the nucleus, thus inducing the transcription of their target genes. In the Wnt/PCP pathway, activated Fzd-co-receptor-Dvl complex induces the activation of Rho family small GTPases, including RhoA, Rac and Cdc42 [11]. Cdc42 and Rac promote the c-Jun N-terminal kinase (JNK) signaling, resulting into the activating protein-1 (AP-1) complex activation [12]. On the other hand, RhoA induces the activation of Rho-associated kinase (ROCK) [13]. These pathways result in the regulation of cell motility and polarity [14,15].

## 3. Wnt Pathway, Inflammation and Carcinogenesis

Chronic inflammation is a widely recognized risk factor for several skin cancers, especially cutaneous squamous cell carcinoma (cSCC) [16]. The role of Wnt signaling in inflammation is complex, because it includes both anti- and proinflammatory functions [17]. Particularly, Wnt signaling would exert its immunomodulation acting on important inflammatory cytokines, such as the nuclear factor kappa B (NF-kB) and its target genes *IL6*, *IL8* and *TNFA*, encoding tumor necrosis factor-α (TNF-α) [17,18,19,20]. Wnt signaling alterations are involved in the pathogenesis of cutaneous chronic inflammatory diseases, as well as those affecting the skin, such as psoriasis, and autoimmune diseases [18,21,22]. In psoriasis, non-canonical Wnt pathway increases keratinocyte proliferation and secretion of pro-inflammatory cytokines, such as TNF-α, interleukin-12 (IL-12) and (IL-23) [23]. Moreover, it is important to mention the role of canonical and non-canonical Wnt pathways in regulation of immune cells, particularly T cells and dendritic cells. The non-canonical Wnt pathway promotes migration of T cells via the CXC chemokine ligand-12 (CXCL12)–CXC chemokine receptor-4 (CXCR4) signaling [24]. Nevertheless, Wnt5a, via non-canonical Ca(2+)/CaMKII /NF-κB signaling, may also induce an abnormal phenotype of dendritic cells, which show an altered response to Toll-like receptor (TLR) ligands [25]. This phenotype is tolerogenic and characterized by increased production of IL-10 [25]. On the other hand, the canonical Wnt pathway represses the T regulatory cells function, promoting autoimmune response [26]. The imbalance of immune responses mediated by Wnt pathways may result in psoriatic inflammation but also in autoimmune disease, such as systemic lupus erythematosus (SLE) or systemic sclerosis (SSc) [23]. Canonical Wnt pathway is hyperactivated in SLE and plays a role in renal fibrosis of lupus nephritis, promoting the differentiation of T cell in Th17 clones [23]. An increased activation of Wnt signaling is present also in skin fibroblasts of patients with SSc, leading to skin fibrosis [23]. The deficiency of Wnt inhibitory factor-1 (WIF1) in fibroblasts of SSc patients is correlated with an hyperactivation of Wnt/*β*-catenin and, thus, with an increased production of collagen [27]. It is well known that reactive oxygen species (ROS) play an important role in fibrotic processes in SSc [28]. Preventing the accumulation of ROS in cultured SSc patient cells restored WIF-1 expression, thus avoiding collagen accumulation [27]. In conclusion, ROS promote Wnt activation, which contributes to fibrosis [27].

As in cutaneous inflammatory and autoimmune diseases, the Wnt pathway may modulate inflammatory responses in cutaneous cancers and influence tumor microenvironment (TME), thus promoting tumor development [29]. Wnt pathway and cancer are in relation to the complement system, which is involved not only in anti-tumor but also in pro-tumorigenic immune responses [16]. Complement component 3 (C3) and its active form, complement anaphylatoxin (C3a), may promote the epithelial-mesenchymal transition (EMT), thus giving invasive properties by decreasing E-cadherin expression [30]. C3a effects are also mediated by the transcription factor Twist [30], which induces the EMT through β-catenin [31]. In vitro experiments showed that exposure to C3 induces cyclin D1 and metalloproteases up-regulation which promote proliferation and migration of cSCC cells [32]. Moreover, C3a exposure stimulates β-catenin and Sox-2, a transcription factor which regulates cell stemness and induces pluripotent stem cells; in fact, C3a receptor silencing led to the down-regulation of β-catenin and Sox-2 with the consequent decrease of tumor volume [32]. The most important contribution of Wnt signaling to pro-tumorigenic inflammation is not limited to the complement system but also concerns T cells function: the thymic stromal lymphopoietin (TSLP) is a pro-inflammatory cytokine which antagonizes skin carcinogenesis by regulating the functions of CD8 and CD4 T cells [33]. TSLP-deficient mice show pro-tumorigenic inflammation and a predisposition to develop tumor growth through Wnt/β-catenin signaling involvement [33]. In fact, Wnt signaling may influence cancer immune-surveillance and facilitate a tumor immune escape, by reducing the recruitment of dendritic cells and by impairing the activity of T regulatory cells and cytotoxic T lymphocytes [34,35]. Tumors responding better to immunotherapy are characterized by a T cell-inflamed tumor microenvironment (TME), namely infiltrating antigen-specific T cells [35]. The TME may influence the growth and progression of tumors [35]. The activation of Wnt/β-catenin pathway prevent T-cell infiltration and activity in TME [35]. This results in progression of tumors and resistance to immunotherapy [35]. There are three main mechanisms underlying tumor immune escape by the Wnt/β-catenin pathway. First, the canonical Wnt pathway induces the activating transcription factor 3 (ATF3), thus inhibiting the transcription of *C-C* motif chemokine ligand 4 (CCL4) in a mouse model. This results in defective infiltration and activation of dendritic cells and T-cells. Another mechanism involved in immune escape concerns the crosstalk between tumor cells and tumor-associated macrophages (TAMs). Tumor cells may stimulate IL-1β production in TAMs via Snail, a transcription factor of canonical Wnt pathway. In turn, IL-β may increase the availability of β-catenin in tumor cells. Finally, canonical Wnt pathway may influence the activity of T regulatory cells, enhancing their survival [35]. A combination therapy based on the use of immune check point inhibitors plus Wnt signaling inhibitors may improve antitumor immunity [36].

As more broadly described in the next paragraphs, several components of Wnt signaling pathways are involved in pathogenesis of NMSCs, both basal cell carcinoma and squamous cell carcinoma. Table 1 summarizes these components.

## 4. Wnt and Basal Cell Carcinoma

Basal cell carcinoma (BCC) is the most frequent skin tumor. Sun exposure is the most important risk factor for BCC. In particular, intermittent, intense sun exposure appears more associated to BCC than cumulative, long-term UV exposure [37]. Other risk factors include light skin color, tendency to burn rather than tan, exposure to ionizing radiation, immunosuppression and predisposing genetic syndromes, such as nevoid basal cell carcinoma syndrome or xeroderma pigmentosum [37]. The origin of BCC cells is controversial and may be derive from basal keratinocytes of the interfollicular epidermis or of the hair follicle [37]. The pathogenesis of BCC is strictly related to an aberrant Sonic Hedgehog (SHH) signaling [38]. SHH signaling is activated following glycoprotein SHH binding to the transmembrane patched receptor (PTCH). It is essential in embryonic development and organogenesis, but it is also involved in tumorigenesis. In physiological conditions, the lack of SHH ligand allows PTCH to inhibit smoothened (Smo), which is a transmembrane G protein-coupled receptor involved in the regulation of the activity of suppressor of fused (SUFU), a negative regulator of the SHH pathway [39]. Generally, SUFU inactivates Glioma associated oncogene homolog 1 (GLI) transcription factors after its binding; when Smo is active, it inhibits the binding of SUFU to GLI. Thus, GLI transcription factors may initiate the transcription of target genes, which regulate cell growth and proliferation [40]. An overexpression of SHH pathway leads to carcinogenesis and may contribute to the development of BCC. Several mutations of SHH pathway components, including loss of PTCH or overexpression of Smo or Gli, have been identified as responsible for BCC development [38,41,42]. Nevertheless, a crosstalk between SHH- and Wnt- pathways has been demonstrated and may be involved in BCC with a critical role played by β-catenin [43] (Figure 2).

BCCs are characterized by nuclear localization of β-catenin [44] and its overexpression may be related to a more aggressive cancer [45]. In addition to SHH, BCCs show different mutations of the components of Wnt/β-catenin pathway, particularly in the exon 3 of the gene encoding for β-catenin [46,47]. Epigenetic studies demonstrated the hyper-methylation of several genes associated with SHH and Wnt signaling pathways, resulting in a constitutive activation of these pathways in BCC [48]. As previously mentioned, β-catenin signaling is regulated by SUFU and GLI, two key components of SHH signaling. In particular, SUFU is a negative regulator of β-catenin, thus acting as tumor suppressor [49], whereas GLI promotes β-catenin activation, thus acting as oncogenic factor [50]. In fact, GLI induces Snail, a repressor of E-cadherin, thus inhibiting the complex E-cadherin/β-catenin in the cell surface so that β-catenin moves into the nucleus and promotes Wnt target genes transcription [39]. Additionally, the Wnt inhibitory factor-1 (WIF1) may regulate the SHH signaling, thus efficiently inhibiting SHH signaling [51]. These observations suggest that WIF1 may play an important role in the development of BCC, being a tumor suppressor of both SHH and Wnt signaling pathways. The significant role of Wnt signaling in BCC is suggested by the presence of mechanisms of resistance, mediated by Wnt pathways, to vismodegib, an SHH inhibitor widely used for the treatment of aggressive BCC [52]. In particular, Wnt pathway activation as well as the permissive chromatin state allowed murine BCC cells to switch their identity and survive to SHH inhibition [52]. The combined treatment with vismodegib and the anti-LRP6 antibody, which is the most important co-receptor of Wnt in BCC, led to a 33% of tumor decrease in a murine model compared to the monotherapy with vismodegib: the combined therapy was effective also in the delay of tumor regrowth after therapy discontinuation [52]. On the basis of this evidence, Wnt signaling might be targeted in the clinical scenario of aggressive BCCs.

## 5. Wnt and Keratinocyte Carcinomas: From Actinic Keratosis to Cutaneous Squamous Cell Carcinoma

Cutaneous squamous cell carcinoma (cSCC) is the second most common tumor after BCC and the second most frequent cause of death from skin cancer after melanoma [53]. CSCC arises from the malignant proliferation of epidermal keratinocytes and actinic keratosis is the premalignant lesion which leads to invasive cSCC. Risk factors for cSCC include male sex, older age, fair skin, human papillomavirus infection, immunosuppression, solid-organ transplant or chronic lymphocytic leukemia. Nevertheless, the most important risk factor for development of cSCC is chronic sun exposure [53]. Actinic keratosis (AK), Bowen’s disease (BD) (carcinoma in situ) and invasive cutaneous squamous cell carcinoma (cSCC) represent the more aggressive expression of sun damage. Ultraviolet radiation (UVR) is the most important risk factor for the development of this kind of lesions and cumulative exposure leads to genetic alterations which firstly induce subclinical atypia and then progress to keratinocyte neoplasms [54]. One of the most common UVR-related mutations concerns tumor protein 53 gene (TP53) encoding p53, a tumor suppressor essential in DNA stability control and cell cycle arrest after DNA damage [55]. TP53 mutations have been found in cSCC, AKs and sun-damaged skin but also in normal sun-exposed skin, suggesting that this mutation early appears in the development of keratinocyte carcinomas [55]. These mutations reduce the defense systems and allow the accumulation of UV-related additional mutations in a progressive process from photodamage to carcinoma [56]. The concept that the cSCC is the last consequence of a series of additional mutations starting from AKs allows to understand why Wnt signaling alterations are detected in precancerous lesions. Additional mutations present in cSCC concerns oncogenes, such as Ras, or tumor suppressors, such as NOTCH or CDKN2A [53]. CDKN2A, a locus encoding the tumor suppressor p16, is mutated in 24–45% of sporadic cSCCs [57]. P16 is a protein which shares with p53 the function of regulating cycle cell [58]; nevertheless, p16 is also overexpressed in precancerous lesions, thus inducing dysplasia [58]. Moreover, p16 promotes the development of early premalignant lesions through Wnt signaling stimulation, which plays an important role in the pathogenesis of cSCC and in the progression of precancerous lesions to aggressive tumors [58]. Genetic studies demonstrated that several components of Wnt pathways were overexpressed in cSCC cells compared to normal skin [59,60]. Both canonical and non-canonical Wnt pathways would be involved in cSCC pathogenesis. For this reason, the role of Wnt in the pathogenesis of cSCC will be described, firstly with β-catenin and canonical Wnt pathway involvement and then with the non-canonical Wnt pathway contribution.

### 5.1. Canonical Wnt Pathway in Pathogenesis of Cutaneous SCC

β-catenin expression gradually modifies from AKs to cSCC. As mentioned above, β-catenin in normal epidermal tissues is located in the cell surface, whereas in pathological conditions, it moves to cytoplasm or nucleus. β-catenin expression on cell membrane shows a gradual down-regulation in AKs, BD and cSCC, while nuclear or cytoplasmatic expression results progressively increased. An immunohistochemical study showed that nuclear or cytoplasmatic expression rates of β-catenin were 56.25%, 91.67% and 96.00% for AKs, BD and cSCC, respectively [61]. One of most important contributions of β-catenin in the pathogenesis of cSCC would be related to its ability to sustain cutaneous cancer stem cells (CSCs) [62]. These cells are phenotypically and functionally similar to normal bulge skin stem cells and are necessary for long-term tumor growth and invasion, as well as being responsible for tumor recurrences [62]. β-catenin would be responsible for CSCs phenotype through activation of several genes, including *c-myc*, widely known for its role in tumorigenesis [62]. Additionally, c-myc expression is progressively positive from normal skin to cSCC, such as β-catenin. In fact, staining of c-myc is negative in normal skin, negative or faint in Bowen’s disease and strongly positive in cSCC [63]. Several studies confirmed that the depletion of β-catenin gene in SCC cells leads to a significant reduction of tumor volume and colony-forming potential of SCC cells [62,64,65]. In addition to TP53, one of most frequent mutations of SCC concerns HRAS, encoding for a protein involved in intracellular signal transduction pathways [66]. Depletion of β-catenin would also be able to impair Hras-dependent tumorigenesis [64]. The tumorigenic effect of β-catenin is also due to its influence on regulation of cell cycle. In particular, β-catenin translocation into the nucleus may induce the transduction of genes essential for cell cycle progression, such as cyclin D1 and cell division cycle 20 (CDC20) [67,68,69] (Figure 3).

CDC20 is a regulator of cell cycle, promoting the progression from metaphase to anaphase, and plays a pivotal role in the development of cSCC, in fact its expression is low in normal skin and higher in cSCC cells, as well as in AK. This evidence suggests that CDC20 is involved in the early events of cSCC onset and in the progression from cSCC in situ to invasive cSCC. On the other hand, CDC20 depletion arrests cell growth and progression of tumor, inducing apoptosis, through Wnt/β-catenin signaling inhibition. Moreover, CDC20 plays an important role in the translocation of β-catenin from cell membrane to the nucleus; thus, CDC20 silencing would lead to a reduced expression of β-catenin and c-myc [69].

Additionally, human papillomavirus (HPV)-related cSCC would be driven by Wnt/β-catenin signaling activation, in particular in immunocompromised patients [70]. Both Wnts and Porcupine were increased in a mouse model of HPV-driven cSCC. Porcupine is an enzyme of endoplasmic reticulum, which acylates Wnt ligands, thus inducing their secretion. The use of LGK974, an antagonist of Porcuspine, in mice with HPV-driven cSCC resulted in a reduced secretion of Wnt ligands as well as in a contained cSCC progression [71]; therefore, Porcuspine could be considered a target for the treatment of cSCC and would control Wnt ligands secretion [70,71].

Different studies found that Wnt antagonists were down-regulated in cSCC. Two classes of extracellular Wnt antagonists are recognized according to the mechanisms of action: the first class, including Cerberus, Wnt inhibitory factor-1 (WIF-1) and secreted Frizzled-related proteins (SFRPs), binds to Wnt ligands; the second class, consisting of Dickkopf (Dkk) proteins, binds to Lrp5-6, which are part of Wnt receptor complex [72]. SFRPs may inhibit both canonical and non-canonical signaling pathways, whereas Dkks antagonize canonical pathway only [73]. Several SFRPs promoters are hypermethylated in cSCC [74] and expression rates of Dkk1 progressively decrease from AKs to cSCC (50.00%, 12.50% and 8.00% for AKs, BD and cSCC, respectively) [60]. This gradual down-regulation of Dkk1 suggests that it is a key suppressor of epidermal cancer progression [60]. Additionally, Dkk3 expression seems to have a progressive alteration from normal epitelium to cSCC [75]: in samples obtained from normal skin, staining of Dkk3 did not show alterations and was negative or very low in cSCC in situ or cSCC [76]. Focally positive immunostaining was observed in AKs [76]. As described above, the stimulation of Wnt/β-catenin pathway results in Lef/Tcf transcription proteins activation, which was also studied in a mouse model of skin carcinogenesis: SCC was reduced by β-catenin/Tcf signaling inhibition [77]. Recently, the relationship between Wnt/β-catenin modulation and some transcription factors, such as Slug, Twist or Snail, was studied in order to evaluate the EMT role in cSCC; in particular, the gene ZFP36, encoding the Zinc-finger RNA-binding protein Tristetraprolin (TTP), was studied [78]. TTP is considered a tumor suppressor, thanks to its ability in arresting tumor growth and reducing the antitumor immune response of CD8+ lymphocytes. TPP may revert EMT through the down-regulation of the transcription factors Slug, Twist and Snail, which may be induced by Wnt/β-catenin pathway and ZFP36 gene. Therefore, the Wnt/β-catenin pathway may further promote the EMT in cSCC [78].

The canonical Wnt pathway is involved in the epithelial–mesenchymal transition (EMT), which is essential for the invasion and progression of tumors [79]. The connections between E-cadherin and β-catenin play a central role in this process. As mentioned above, E-cadherin mediates cell adhesion and stabilizes the architecture of tissues [80]. Suppression of E-cadherin leads to alterations of cellular architecture, with loss of cellular junctions and tumoral invasion [81]. When E-cadherin is lost, β-catenin moves to the nucleus and induces the transcription of target genes [80]. Among these, ADAM10 is a metalloprotease that reduces cell adhesion and promotes cell migration, in addition to induce β-catenin translocation to the nucleus [80]. Wnt/β-catenin target genes include other factors involved in destabilization of cell junction, such as Slug [82] or Twist [83], as well as pro-invasive factors, such as laminin-5γ2 [84], which promotes the motility of epithelial cells and loss of polarity [80].

### 5.2. Non-Canonical Wnt Pathway in Pathogenesis of Cutaneous SCC

The non-canonical Wnt pathway may be involved in the pathogenesis of cSCC, although its role seems to be negligible compared to that described for the canonical pathway (Figure 4).

The non-canonical Wnt pathway may influence SCC development through EMT dysregulation as well as survival and proliferation of cancer cells. For this reason, the non-canonical Wnt pathway may also be considered a possible target to stop tumor progression. An experimental study demonstrated that TAM67 (dominant-negative c-Jun), an inhibitor of the transcription factor AP-1, reduced tumor progression not only through the modulation of the expression of genes involved in inflammation, invasion and metastasis, but also through Wnt5a reduction, whose role in the maintenance of tumor phenotype was previously observed. In fact, Wnt5a knockdown led to inhibition of Signal transducer and activator of transcription 3 (STAT3) target genes, including metalloprotease 3 or cyclin D1, thus suppressing cSCC growth [85]. The inhibition of Gsk3β by Dvl, consequent to the canonical pathway activation, induces an overexpression of transcription factors of Snail [86,87], which inhibits E-cadherin, inducing proliferation and invasion [61]. Moreover, Snail may activate the non-canonical Wnt pathway, inducing Wnt5a and Ror2 expression. The Wnt5a/Ror2 signaling promotes cell migration and tumor invasion through up-regulation of metalloproteases in epidermoid carcinoma cells [88]. Therefore, both canonical and non-canonical Wnt pathway are linked by a vicious circle, induced by Snail, which causes the self-perpetuation of EMT in different cancers, including skin carcinomas. However, Wnt pathways are not the only cell signaling which plays a significant role in EMT; surely, one of the most important EMT regulators is the Transforming Growth Factor–β (TGF-β) [89], in fact TGF-β and Wnt pathways would share several points of crosstalk [90]. For instance, Lef1/Tcf is the transcription factor involved in Wnt/β-catenin pathway, but may be also activated by SMAD proteins, as a consequence of TGF-β modulation [90,91]. Additionally, CUTL-1, a target of TGF-β involved in tumor invasion, may transcriptionally up-regulate Wnt5a, encoding the ligand of non-canonical Wnt pathway [90,92]. In vitro migration assays showed that a Wnt5a gradient may increase cell motility towards adjacent tissue although homogeneous concentration of Wnt5a inhibits chemotactic migration, thus regulating cancer progression [93]. 

## 6. Conclusions

Wnt pathways play a significant role in the pathogenesis of NMSCs through the activation of inflammatory processes involved in carcinogenesis as well as stimulation of cancer cell proliferation and invasion. Moreover, Wnt signaling activation sustains mechanisms of resistance to SHH pathway inhibitors, which currently represent the most advanced therapy for locally advanced or metastatic BCC. Both canonical and non-canonical Wnt pathways are involved in cSCC pathogenesis, sometime synergistically. The activation of the canonical Wnt pathway may contribute to the progression from precancerous to malignant lesions, thus sustaining cutaneous cancer stem cells, and participates with the non-canonical pathway to epithelial–mesenchymal transition. Therefore, the Wnt pathway may be considered a possible therapeutic target for the management of NMSCs in order to reduce tumor burden and improve immune checkpoint inhibitors effects in patients with BCC, also slowing the progression from AKs to cSCC and reducing the recurrences and metastatic invasion of cSCC. For this reason, Wnt signaling still represents an interesting topic to study in different research areas, such as oncology, pharmacology as well as dermatology, in order to find new possible therapeutic targets for diseases that are difficult to treat, including NMSCs.

## Figures and Tables

**Figure 1 ijms-24-01575-f001:**
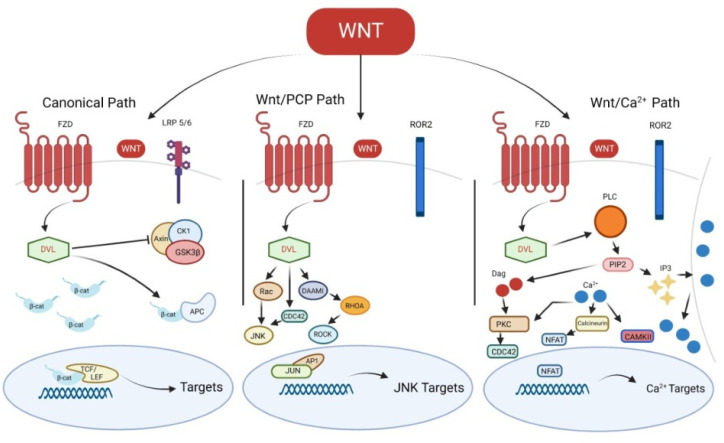
Canonical and non-canonical Wnt signaling pathways. FZD, Frizzled; LRP 5/6, low-density lipoprotein receptor-related protein; DVL, Disheveled; CK1, casein kinase 1α; GSK3β, glycogen synthase kinase 3β; β-cat, β-catenin; APC, adenomatous polyposis coli; TCF/LEF, T-cell factor/lymphoid-enhancing factor; Wnt/PCP, Wnt/Planar cell polarity; ROR2, receptor tyrosine kinase-like orphan receptor 2; JNK, c-Jun N-terminal kinase; DAAMI, Disheveled Associated Activator Of Morphogenesis 1; ROCK, Rho-associated kinase; AP-1, the activating protein-1; PLC, phospholipase C γ; DAG, diacylglycerol; PIP2, phosphatidylinositol 4,5-bisphosphate; IP3, inositol 1,4,5-triphosphate; PKC, protein kinase C; CAMKII, Ca^2+^-calmodulin dependent kinase II; NFAT, nuclear factor of activated T-cells.

**Figure 2 ijms-24-01575-f002:**
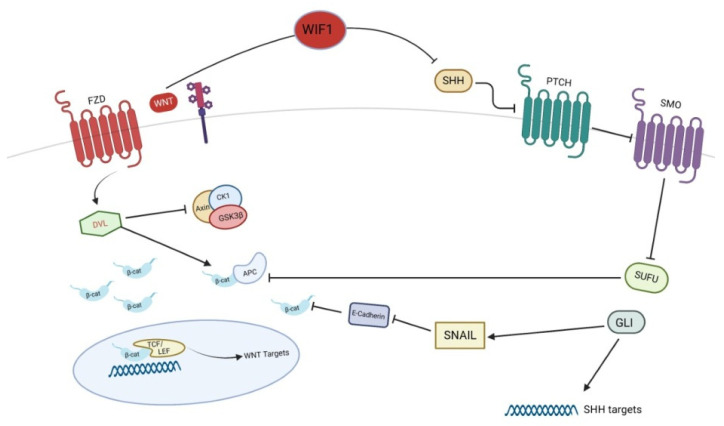
Crosstalk between SHH and Wnt pathways in pathogenesis of basal cell carcinoma. WIF-1, Wnt inhibitory factor-1; SHH, Sonic Hedgehog; PTCH, patched; SMO, smoothened; SUFU, suppressor of fused; GLI, Glioma associated oncogene homolog 1.

**Figure 3 ijms-24-01575-f003:**
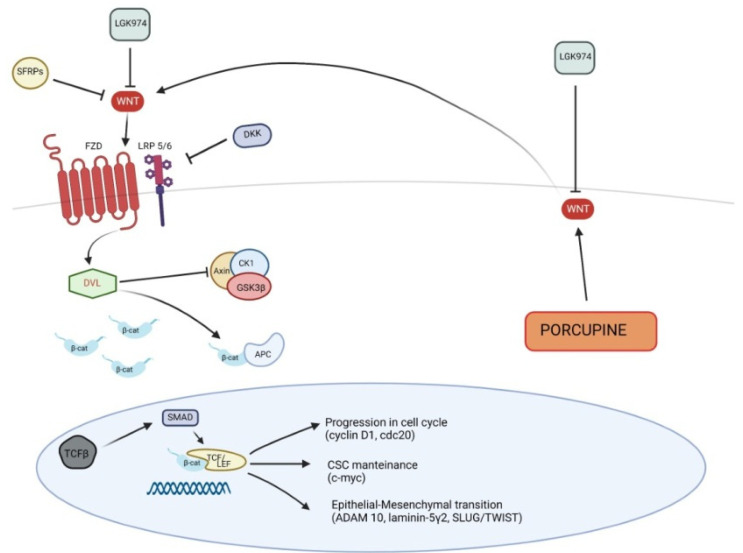
Canonical Wnt signaling pathway in pathogenesis of squamous cell carcinoma. SFRPs, secreted Frizzled-related proteins; DKK, Dickkopf; TGF-β, Transforming Growth Factor-β.

**Figure 4 ijms-24-01575-f004:**
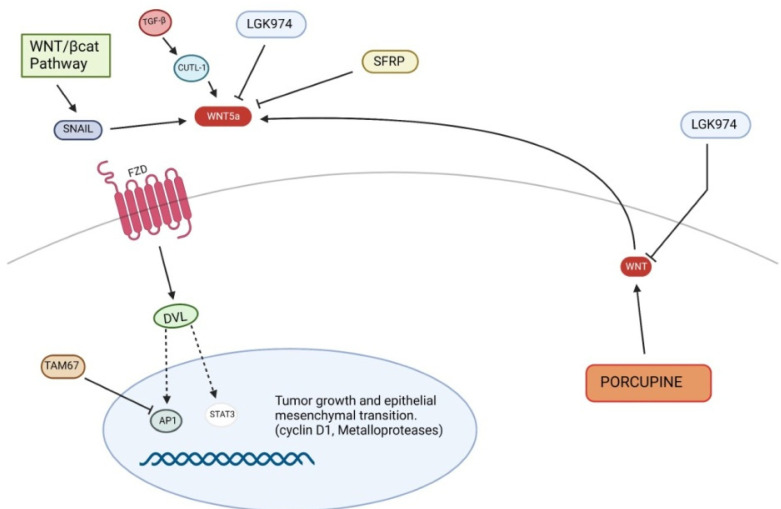
Non-canonical Wnt signaling pathway in pathogenesis of squamous cell carcinoma. STAT3, Signal transducer and activator of transcription 3.

**Table 1 ijms-24-01575-t001:** Canonical and non-canonical Wnt signaling elements involved in pathogenesis of NMSCs.

	Element	Effect
**Basal cell carcinoma**		
** *Up-regulated* **	GLI	promotion of β-catenin activation
** *Down-regulated* **	SUFU	negative regulation of β-catenin
	WIF1	inhibition of SHH signaling
**Canonical Wnt pathway in SCC**		
** *Up-regulated* **	β-catenin	overexpression of canonical Wnt pathway
	Porcuspine	secretion of Wnt from endoplasmic reticulum
	Lef/Tcf	transcription of Wnt target genes
	c-myc	maintenance of cutaneous CSCs
	TGF-β	activation of Lef/Tcf
** *Down-regulated* **	SFRPs	inhibition of Wnt ligands
	WIF1	inhibition of Wnt ligands
	Dkk1-3	inhibition of Wnt receptor complex
**Non-canonical Wnt pathway in SCC**		
** *Up-regulated* **	STAT3	transcription of Wnt target genes
	Snail	inhibition of E-cadherin and promotion of EMT
	CUTL-1	up-regulation of Wnt5a
** *Down-regulated* **	SFRPs	inhibition of Wnt ligands

GLI: Glioma associated oncogene homolog 1; SUFU: suppressor of fused; WIF1: Wnt inhibitory factor-1; Lef/Tcf: T-cell factor/lymphoid-enhancing factor; CSCs: cancer stem cells; TGF-β: Transforming Growth Factor–β; SFRPs: secreted Frizzled-related proteins; Dkk: Dickkopf; STAT3: Signal transducer and activator of transcription 3; EMT: epithelial-mesenchymal transition.

## Data Availability

Not applicable.

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
