# Peer review of "Wnt Signaling Pathways: From Inflammation to Non-Melanoma Skin Cancers"

_ijms, 2023, doi:10.3390/ijms24021575_

Round 1
Reviewer 1 Report
The authors submitted a well-written review article focusing on Wnt signaling pathways from inflammation to non-melanoma skin cancers. The quality/resolution of each Figure is very poor and difficult to read. However, there are several points in the manuscript that they should revise.
1. It is recommended to authors that clearly explain the canonical and non-canonical Wnt signaling elements that are involved in NMSCs, such as Wnt5a and Axin2, in the form of a table so that the reader can understand which Wnt elements are upregulated or downregulated, particularly in skin cancers.
2. The authors mentioned about the “Wnt pathway, inflammation and carcinogenesis”. Wnt signaling alterations are involved in the pathogenesis of chronic inflammatory diseases, also those affecting skin, such as psoriasis.
3. It is suggested that specific details about the factors that cause inflammatory skin diseases such as psoriasis and SLE that may be recognized, as skin cancers be included. Because there are several innate immune signaling pathways, such as toll-like receptor 7/9 (TLR7/9), they are thought to be a major factor in inflammation, psoriasis, and SLE. As a result, it is preferable to highlight those Wnt signaling factors that are responsible for the aforementioned inflammatory diseases (psoriasis, SLE, etc.) and then recognize them as skin cancers.
Author Response
Dear Reviewer,
We would like to thank you for your helpful comments and encouragement. Please find below a point-to-point short response to your questions. We have modified the text, highlighting the changes in red.
Thank you,
Mario Vaccaro (on behalf of co-authors)
Response to Reviewer 1.
The quality/resolution of each Figure is very poor and difficult to read.
A: We modified the figures to improve their quality.
- It is recommended to authors that clearly explain the canonical and non-canonical Wnt signaling elements that are involved in NMSCs, such as Wnt5a and Axin2, in the form of a table so that the reader can understand which Wnt elements are upregulated or downregulated, particularly in skin cancers.
1 A: We added a summary table to the text.
- The authors mentioned about the “Wnt pathway, inflammation and carcinogenesis”. Wnt signaling alterations are involved in the pathogenesis of chronic inflammatory diseases, also those affecting skin, such as psoriasis.
- It is suggested that specific details about the factors that cause inflammatory skin diseases such as psoriasis and SLE that may be recognized, as skin cancers be included. Because there are several innate immune signaling pathways, such as toll-like receptor 7/9 (TLR7/9), they are thought to be a major factor in inflammation, psoriasis, and SLE. As a result, it is preferable to highlight those Wnt signaling factors that are responsible for the aforementioned inflammatory diseases (psoriasis, SLE, etc.) and then recognize them as skin cancers.
2-3 A: We added an explanation on the role of Wnt signaling in cutaneous inflammatory and autoimmune diseases, such as psoriasis, SLE and systemic sclerosis.
Reviewer 2 Report
Wnt Signaling Pathways: from Inflammation to Non-Melanoma Skin Cancers
The authors Bartolomeo et al present the role of Wnt pathway and address various aspects of the pathway and its involvement in cancer/s (NMSCs) and this article falls within the scope of MDPI-IJMS.
Please, see the comments below to improve this paper:
Firstly, all the figures (1-4) look hazy and not crisp. Please, look into the figures and improve the resolution or please redraw.
Line 13: How many Wnt Signaling Pathways are there? Even the title says Wnt Signaling Pathways?
Line 31: “pro-tumorigenic immunomodulation”- very less of this was covered in the entire paper. It would be more useful for the readers if this part is covered a bit more.
Line 33: not pathways? One pathway which has been further categorized into subtypes.
Line 84: With respect to Lines 13 and 33- Continuation: Why don’t you mention here as pathways?
Line 111: is activated.
Line 300: Throughout the paper, the authors discuss the role of Wnt pathway in NMSCs but again in this line they use the word “may”? Please, conclude based on the evidence provided.
Line 304: for locally- Please give space between two words
Line 315: difficult to treat not treatment.

Author Response
Dear Reviewer,
We would like to thank you for your helpful comments and encouragement. Please find below a point-to-point short response to your questions. We have modified the text, highlighting the changes in red.
Thank you,
Mario Vaccaro (on behalf of co-authors)
Response to Reviewer 2.
Line 13: How many Wnt Signaling Pathways are there? Even the title says Wnt Signaling Pathways?
A: The pathways are two, canonical and non-canonical. We specified that in the abstract.
Line 31: “pro-tumorigenic immunomodulation”- very less of this was covered in the entire paper. It would be more useful for the readers if this part is covered a bit more.
A: We added more explanations in the text.
Line 33: not pathways? One pathway which has been further categorized into subtypes.
A: We changed the title of paragraph in “Wnt signaling”
Line 84: With respect to Lines 13 and 33- Continuation: Why don’t you mention here as pathways?
A: It was a typo. We modified it.
Line 111: is activated.
A: We corrected it.
Line 300: Throughout the paper, the authors discuss the role of Wnt pathway in NMSCs but again in this line they use the word “may”? Please, conclude based on the evidence provided.
A: We corrected the paragraph.
Line 304: for locally- Please give space between two words
A: Ok, thanks!
Line 315: difficult to treat not treatment.
A: Ok, thanks!
Reviewer 3 Report
1. The authors should add some literature about the epidemiology, pathogenesis and pathophysiology of Non-Melanoma skin cancer.
2. The author should improve the quality of figures.
Author Response
Dear Reviewer,
We would like to thank you for your helpful comments and encouragement. Please find below a point-to-point short response to your questions. We have modified the text, highlighting the changes in red.
Thank you,
Mario Vaccaro (on behalf of co-authors)
Response to Reviewer 3.
- The authors should add some literature about the epidemiology, pathogenesis and pathophysiology of Non-Melanoma skin cancer.
A: We added information on BCC and SCC in their respective chapters.
- The author should improve the quality of figures.
A: We modified the figures to improve their quality.